

# Water levels of the Mekong River Basin based on CryoSat-2 SAR data classification

Eva Boergens[1], Karina Nielsen[2], Ole B. Andersen[2], Denise Dettmering[1], and Florian Seitz[1]

[1]Deutsches Geodätisches Forschungsinstitut der Technischen Universität München, Arcisstr. 21, 80333 München
[2]Div. of Geodesy, DTU Space, National Space Institute, Kgs. Lyngby, Denmark

*Correspondence to:* Eva Boergens (eva.boergens@tum.de)

**Abstract.** In this study we use CryoSat-2 SAR (Delay-Doppler Synthetic Aperture Radar) data over the Mekong River Basin to estimate water levels. Smaller inland waters can be observed with CryoSat-2 data with a higher accuracy compared to the classical radar altimeters due to the increased along track resolution of SAR and the smaller footprint. However, even with this SAR data the estimation of water levels over smaller (width less than 500 m) is still challenging as only very few consecutive observations over the water body are present. The usage of land-water-masks for target identification tends to fail as the river becomes smaller. Therefore, we developed a classification to divide the observations into water and land observations based solely on the observations.

The classification is done with an unsupervised classification algorithm, and it is based on features derived from the SAR and RIP (Range Integrated Power) waveforms. After the classification, classes representing water and land are identified. The measurements classified as water are used in a next step to estimate water levels for each crossing over the Mekong River. The resulting water levels are validated and compared to gauge data, Envisat data and CryoSat-2 water levels derived with a land-water mask. The CryoSat-2 classified water levels perform better than results based on the land-water-mask and Envisat. Especially, in the smaller upstream regions the improvements of the classification approach for CryoSat-2 are evident.

## 1 Introduction

The water of rivers is vital for humans but poses a threat at the same time. Rivers are crucial as a suppliers of water for irrigation and fresh water for drinking. However, floods can destroy crops, settlements, and infrastructure. For this reason, it is essential to monitor the water level of river systems. An increasing number of in situ gauges have been derelicted since the 1980s (Center, 2013), or the data is not publicly available. It is therefore increasingly important to measure river water level with satellite altimetry.

In recent years many studies were published that apply satellite altimetry over rivers of various sizes (e.g. Birkett, 1998; Santos da Silva et al., 2010; Schwatke et al., 2015; Boergens et al., 2016b; Frappart et al., 2006; Biancamaria et al., 2016). All of the aforementioned studies use pulse-limited altimetry data. CryoSat-2, launched in 2010, is the first satellite carrying a Delay-Doppler altimeter (Raney, 1998). The altimeter operates in three measuring modes: the classical pulse-limited Low Resolution



(LR) mode, the Delay-Doppler Synthetic Aperture Radar (SAR) mode, and the SAR Interferometric (SARin) mode. The modes of the altimeter are governed by a geographical mode mask (https://earth.esa.int/web/guest/-/geographical-mode-mask-7107).

Compared to conventional radar altimeters, Delay-Doppler measurements have a higher along track resolution and a smaller footprint. This improves the observation of water levels of inland water bodies like lakes (e. g. Nielsen et al., 2015; Kleinheren-
brink et al., 2015; Göttl et al., 2016) or rivers (e. g. Villadsen et al., 2015; Bercher et al., 2013). The advantage of SAR altimetry observations are especially useful for measuring smaller inland waters like rivers. However, CryoSat-2 has a long repeat time of 369 days compared to 35 days of Envisat and SARAL, and 10 days for Topex/Poseidon, Jason-1 and Jason-2. This restricts the estimation of meaningful water level time series over rivers or lakes, if not enough different tracks cross the water body. The advantage of the long repeat time is the very dense spatial distribution of observations.

To derive water levels from lakes or rivers it is necessary to identify the water returns of the altimeter. This can be done by applying a land-water-mask such as the mask provided by the World Wildlife Fund (https://www.worldwildlife.org/pages/global-lakes-and-wetlands-database). Such a mask is constant over time, therefore, it neither accounts for the seasonal variations of the water extent nor inter-annually shifting river and lake banks. These masks are usually not accurate enough for narrow rivers where only a few water measurements are available. Although a high accuracy land-water-mask is provided by
the Mekong River Commission (http://portal.mrcmekong.org/map_service) for our study area of the Mekong River Basin, its accuracy of 30 m is still not sufficient for the smaller and, especially, the small rivers. In the Mekong River Basin the river width varies between 20 m to more than 2 km. The small rivers with a width of less than 100 m are most of the tributaries and the upstream part of the left river bank side main tributaries. The smaller rivers, which are less than 500 m wide, are the main tributaries and the upstream main river. In the downstream reach of the river, before it splits into the delta, the river has a width
of over 2 km (see also Figure 1 for a map of the basin).

To overcome the problems and limitations of land-water-masks, we classify the altimetry data beforehand in water and land observations. For the classical pulse-limited altimeter this has been done successfully for the last decade (e.g Berry et al., 2005; Desai et al., 2015). Even very small water areas in wetlands have been classified successfully with Envisat data by Dettmering et al. (2016). In the classification, the shape of the waveform is used to discriminate between different reflecting surfaces. Also
CryoSat-2 SAR data has been classified based on the SAR waveform before for lakes (Göttl et al., 2016), lakes and rivers (Villadsen et al., 2016), or ice (Armitage and Davidson, 2014). This study takes a step further and uses not only the waveform but also the Range Integrated Power (RIP) for a classification of the altimeter measurements in water and non-water returns over the Mekong River Basin in Southeast Asia. The RIP is only available for Delay-Doppler SAR altimetry and gives additional insight to the reflective surface which the waveform alone could not provide (see Figure 2 for an example and Wingham et al.
30 (2006)).

The unsupervised *k-means* algorithm is used for the classification (MacQueen, 1967) as not enough reliable training data is available for a supervised classification. The *k-means* algorithm is a widely used unsupervised clustering algorithm and has been used for altimetry classification before (e.g. Göttl et al., 2016).

This paper is structured as follows: First, an introduction is given about the study area of the Mekong River Basin in section 2,
afterwards more information of the CryoSat-2 SAR data is given in section 3. The classification and the used features are



described in section 4 followed by an explanation of the water level estimation in section 5. The results and validations are presented and discussed in section 6. The paper ends with the conclusions in section 7.

## 2  Study Area

In this study, the Mekong River Basin in Southeast Asia (China, Myanmar, Thailand, Laos, Cambodia, and Vietnam) is investi-
gated, with focus on the part of the basin south of the Chinese border. Upstream from here, it is not possible to measure the river with satellite altimetry because the river flows through narrow gauges that shadow the altimetric measurements. Downstream, the study area ends by the confluence with the Tonle Sap River from where on the river is under tidal influence. The tributaries, namely the large left bank side tributaries in Laos, are investigated as well. The hydrology of the Mekong Basin is primarily influenced by the precipitation on the Tibetan Plateau and the south-eastern monsoon (Commission, 2005).

The Mekong River and its tributaries flow through different topographic regions (Figure 1). The main river upstream from Vientiane and the left bank tributaries in Laos are surrounded by mountainous areas with steep banks where the rivers have a greater slope and have a width smaller than 500 m or even less than 100 m. Downstream of Vientiane and up to the Mekong Falls the river widens and flows with less slope over the Khorat plateau. Below the Mekong Falls the river is surrounded by seasonal wetlands and widens to more than 1 km. For further processing we defined three overlapping data masks according to these regions (Figure 1).

## 3  Data

In this study we use Delay-Doppler SAR altimeter data measured by CryoSat-2 between 2010 and 2016. CryoSat-2 measures in three different modes, which are set in a geographical mask (https://earth.esa.int/web/guest/-/geographical-mode-mask-7107). The LRM is active mostly over the oceans and the interior of the ice sheets of Antarctica and Greenland, whereas the SAR mode measures over sea ice and other selected regions and SARin focuses mostly on glaciated regions (ESA, 2016). This mask has changed over the life time of the satellite. The entire study area of the Mekong River Basin has only been measured in SAR mode since July 2014 (see Figure 1 for the extent of the SAR mode mask). In SAR mode the along-track foot print size is reduced to 300 m while it remains 10 km in the across-track direction.

Here, we use the CryoSat-2 baseline C SAR Level 1b data provided by ESA GPOD SARvatore (https://gpod.eo.esa.int/) for the period of 2010 to 2016. These data contain the full stack matrix.

The Delay-Doppler SAR altimeter measures a point on the surface several times from different looking angles (Cullen and Wingham, 2002). All these measurements form the multi-look stack data (see Figure 2). For every point 246 single-look waveforms are collected in the stack matrix. In Figure 2, two exemplary stack matrices are presented. The first (a) is measured over the Tonle Sap lake and the second (b) over a smaller river upstream. Each row is a single-look waveform. The integration of this matrix over all single-looks results in the multi-look SAR waveform (in Figure 2 integration over each row of the stack) hereafter referred to as the waveform. The integration over the range bins results in the Range Integrated Power (RIP). In



**Figure 1.** Map of the study area with the regional masks (black areas with different hachures) and the SAR mode mask with their validity period (red boxes).

Figure 2 this corresponds to the integration over the columns. Detailed information on the Delay-Doppler measurements are described in Raney (1998).

Additionally, we use a river polygon which is provided by the Mekong River Commission (http://portal.mrcmekong.org/map_service). The polygon was derived from aerial images and topographic maps. The accuracy of the river mask is ∼ 30 m, but no information about seasonality of the polygon is given.





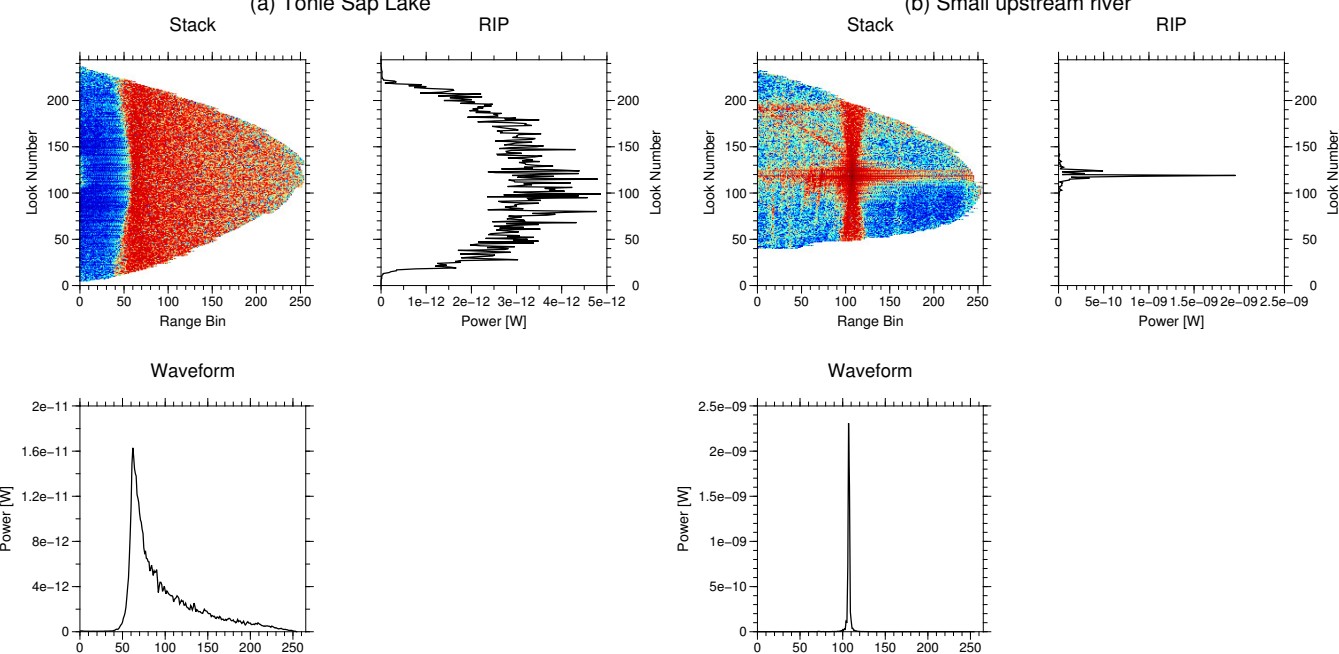

**Figure 2.** Two exemplary stack matrices with their RIP and waveform. The color of the stack plot indicates the power of the signal. The example on the left hand side is measured over the Tonle Sap lake, the one on the right hand side over a smaller upstream river.

## 4 Classification Approach

For the smaller and small rivers in our study area of the Mekong basin no reliable land-water-mask is available. Thus a classification by means of the *k-means* algorithm is performed to extract the water measurements.

The *k-means* algorithm (MacQueen, 1967) is an unsupervised method to cluster the data on the basis of different features.
5 For the land-water classification a set of features derived from both the waveform and the RIP is used which are summerized in Table 1. The features derived from the waveform are the *maximum power*, the *peakiness*, and the *position of the leading edge*. It is well known, that waveforms of water reflections have a higher power than those of land reflections. Smaller, and even more so small, water bodies have a smooth mirror-like surface which can only be measured by signals emitted close to nadir. This leads to a very peaky waveform and RIP with a high power. Following Laxon (1994) the peakiness $\mathbf{p}_{\mathrm{wf}}$ is calculated with

$$10 \quad \mathbf{p}_{\mathrm{wf}} = \frac{\max(\mathrm{wf})}{\sum\limits_{i} \mathrm{wf}_i}, \tag{1}$$

where wf is the waveform and $\mathrm{wf}_i$ the power of the $i^{th}$ bin.

To estimate the relative *position of the leading edge* in the waveform, the waveform is retracked using an Improved Threshold Retracker with a threshold of 50% on the best sub-waveform (Gommenginger et al., 2011). The on-board tracking system always tries to hold the leading edge of the main reflection at the nominal tracking point. This is not always possible and leads





to a deviation of the leading edge from the nominal tracking point. Over wider rivers the tracking system can manage to keep the leading edge close to the tracking point. In Figure 3, left panel, one exemplary waveform with its features *maximum power* and *position of the leading edge* is shown (the *peakiness* cannot be displayed).

Features based on the RIP are the *peakiness* $\mathbf{p}_{\mathrm{RIP}}$, the *standard deviation* $\mathbf{std}_{\mathrm{RIP}}$, the *width*, the *off-center*, and the *symmetry*.

The $\mathbf{std}_{\mathrm{RIP}}$ is a measure of the the difference in the returning power under different looking angles is (see Figure 2). Water reflections over larger water bodies result in a overall smoother RIP than water reflections over smaller water bodies which in turn have a smoother RIP than land reflections. The $\mathbf{std}_{\mathrm{RIP}}$ is

$$\mathbf{std}_{\mathrm{RIP}} = \sqrt{\frac{1}{N}\sum_i \left(\mathrm{RIP}_i - mean(\mathrm{RIP})\right)^2}, \tag{2}$$

where $\mathrm{RIP}_i$ is the $i^{th}$ entry of the RIP and $N$ the number of looks in the RIP, usually 246.

As mentioned before, smaller inland waters with a smooth surface only reflect the signal back to the satellite at near nadir. Therefore the RIP is both very peaky and narrow. The width $\mathbf{w}_{\mathrm{OCOG}}$ is derived by the formula of the OCOG retracker (Gommenginger et al., 2011):

$$\mathbf{w}_{\mathrm{OCOG}} = \frac{{\sum\limits_i \mathrm{RIP}_i^2}^2}{\sum\limits_i \mathrm{RIP}_i^4}. \tag{3}$$

The off-center feature $\mathbf{off}$ describes the deviation of the main reflection from the nadir point. It should be close to zero for

measurements of water, whereas land measurements are more disturbed and often show the maximum return in the lobes. We measure the off-center feature $\mathbf{off}$ as the difference between the middle look of the RIP and the mean point of the RIP which is calculated with the formula of the centre of gravity from the OCOG retracker:

$$\mathbf{off} = \frac{246}{2} - \frac{\sum\limits_i i\,\mathrm{RIP}_i^2}{\sum\limits_i \mathrm{RIP}_i^2}. \tag{4}$$

A positive $\mathbf{off}$ value indicates that the majority of the returning power was detected before the satellite passed the nadir position,

a negative value vice versa.

The last feature is a measure of the symmetry of the RIP $\mathbf{s}$. For an ideal smooth water reflection, like a small lake, the RIP should be perfectly symmetrical. However, for a sloped target, as a river is, the reflection depends on the relative orientation between the satellite and the water surface. The reflection is stronger when the satellite looks on a water surface that is sloped towards it. This effect leads to an unsymmetrical RIP. To quantify this, an unsymmetrical exponential function $\overline{\mathrm{RIP}}$ is fitted to

the RIP with

$$\overline{\mathrm{RIP}_i} = \begin{cases} a\exp\left(\frac{(i-b)^2}{2c_1^2}\right), & \text{if } i < b \\ a\exp\left(-\frac{(i-b)^2}{2c_2^2}\right), & \text{if } i \geqq b. \end{cases} \tag{5}$$




**Table 1.** Features used for the classification

| RIP features | Waveform features |
|---|---|
| Peakiness: $\mathbf{p}_{RIP}$ | $\mathbf{p}_{wf}$ |
| Standard deviation: $\mathbf{std}_{RIP}$ | Maximum power: $\mathbf{max}_{wf}$ |
| OCOG width: $\mathbf{w}_{OCOG}$ | Relative position of leading edge |
| Off-center: $\mathbf{off}$ | |
| Symmetry: $\mathbf{s}$ | |

Here, $a$ is the amplitude of the exponential function, $b$ the look where the function reaches its maximum, and $c_1$ and $c_2$ are the two decay parameters. The *symmetry* feature is then

$$\mathbf{s} = c_1 - c_2. \tag{6}$$

Figure 3, right hand, displays a RIP with the feature $w_{OCOG}$ marked. The off-center feature $off$ is too small to be visible in this example, but the symmetry, or the lack thereof, is clearly shown.

Additional to these eight features, both the whole waveform and the whole RIP are used as features. Each bin is then considered as a single feature. The waveform needs to be shifted so that the leading edge is positioned on the nominal tracking point. Since the features span different orders of magnitude, it is necessary to normalize the feature set.

The *k-means* algorithm is used to cluster the data on the basis of the above features in 20 classes. The number of classes depends on the application and variation in the input features. An estimate for the number of classes can be done with knowledge of the classified data. In our study case, a look at the spatial distribution of the features tells us that only two classes, land and water, are not sufficient as altimeter measurements of land can be very diverse (this holds also for water measurements, but they are less diverse than land). The diversity of the returning waveform and RIP can be explained by the reflective properties of e. g. land, water, vegetation. With this it can be concluded that at least 10 classes are needed. We tested the classification and validated resulting water levels for a several numbers of classes (10, 15, 20,30) and found similar results for all with the results of 20 classes slightly superior.

Each of the clusters is defined by their centroid which is the mean feature of all points in this cluster. New data is then classified by grouping it to the closest centroid. Here, the clustering is done on one randomly drawn third of the data. The residual two third of the data is then classified into the cluster classes. The clustering is not done on the whole data set due to computational efficiency. The repeatability of the clustering and classification will be validated in section 6. After the classification it is determined which classes represent water and land returns, respectively. This was done by visual inspection of the mean waveform and RIP for each class and the locations of the observations in each class related to the land-water-mask (see section 3).

As described in section 2 the Mekong Basin is divided into different regions, upstream, middle and downstream. We classify each of the regions separately as they are too diverse in the reflectivity properties of the water bodies to be classified together.





Additionally, the classification is done only on altimeter data not further away than 20 km from the river polygon due to computational efficiency (the polygon can be seen in Figure 1).

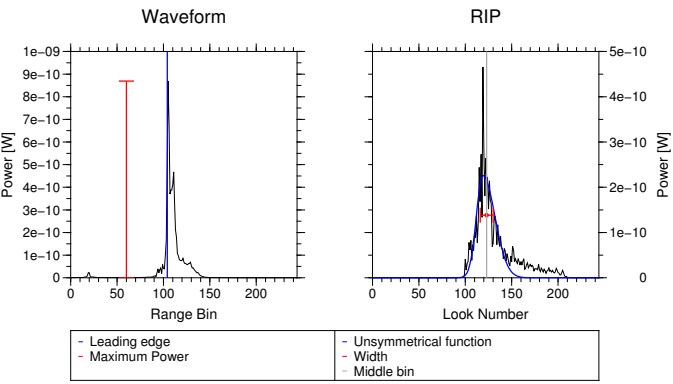

**Figure 3.** One example of a waveform and the corresponding RIP with some of their derived features.

## 5 Water Level Estimation Approach

The classification results in a set of measurements considered as water returns. From these measurements the water level for each crossing is determined in this section.

### 5.1 Altimetric Water Levels

A water level is computed for each crossing of the satellite track with a river in the Mekong River Basin. To locate these crossings a river polygon (see section 3) is used. We apply all measurements less than 5 km away from the river crossing that were classified as water and retrack the SAR waveforms with an Improved Threshold Retracker with 50% threshold (Gommenginger et al., 2011). Instead of using a median or mean over all classified measurements, we search for a horizontal line in the heights, which is assumed to represent the water surface. It is still possible that some of the water classified measurements do not represent the river surface and need to be excluded from the water level computation (across-track of nadir effects or water bodies surrounding the river). These outliers do not necessarily have to be at the margin of the river but can also be located in the middle due to islands or sandbanks in the river. This would restrict the use of an along-track standard deviation of the heights for outlier detection.

To find the line of equal water height, a histogram of the water levels with Doane bins (Doane, 1976) is used. Doane bins are more suitable to small (less than 30) non-normally distributed data than the classical Sturge bins (Sturges, 1926). If a horizontal line is present in the heights, one of the bins is distinctively larger, e.g. contains more observations, than the others and collects the heights of nearly equal water level. The median of the heights in this bin is then taken as water level. If less than 5 height points were classified as water, the median of the heights is taken as the water level. The advantage of this approach is that it is





better suited for rivers wider than 1 km with islands and sandbanks that cause outliers in the heights. However, in many cases our histogram approach and taking the median of all observations deliver similar results.

## 5.2 Outlier Detection

In spite of careful data selection through the classification and in the height retrieval, some retrieved water levels have to be
considered as outliers. To find these outliers we make use of the CryoSat-2 repeat time of 369 days. With the knowledge of the very stable annual signal of the Mekong River one can assume that two measurements of the same CryoSat-2 track 369 days apart should measure a similar height. Based on this, a water level is considered as an outlier if the mean difference to all other heights of the same pass is larger than 7 m. This is only applicable if other water level measurements of the same track exist. Due to the changing mode mask (see section 3) some regions are only measured in the last two years. To overcome this,
a second outlier detection is applied which compares the water level with water levels of other tracks that are close in space and time of the year. To this end, we used all measurements that are less than 10 km away along the river and less then 30 days of the year apart. If the water level is more than 10 m different from the distance weighted mean water level of all these points it is considered as an outlier.

The thresholds for the outlier detection were chosen as a conservative upper bound. It has to be expected to have in average a
water level difference of 40 to 60 cm in five days during the rising water season, but it could be as high as 4 or 5 m (Commision, 2009). Additionally, some inter-annual changes in the flood season can be expected, and the rivers in the Mekong Basin have a median slope of 30 cm/km.

## 5.3 Merging of the overlap regions

From the classification we derive a set of heights for each of the different geographical regions which have a certain overlap (see
Figure 1 and section 2). In this overlap, for the same crossing two water levels were computed, therefore, it has to be decided which height shall be used. To resolve this, we use the distance weighted mean water level of all other water level measurements that are less than 10 km away and less then 30 days of the year apart as in the outlier detection (see subsection 5.2). The water level that is closest to this mean water level is applied. The results of the merging process can also be used for validation of the classification as will be shown in subsubsection 6.3.3.

## 6 Results, Validation and Discussion

We applied the described methodology for the classification and water level determination on CryoSat-2 SAR data in the Mekong River Basin. In this section, both the results of the classification and the water level determination are presented and validated.





## 6.1 Results of the Classification

After the clustering and classification of the CryoSat-2 measurements we select the classes of water returns. In the upstream region we identify three and in the middle region six out of twenty as water classes. In the downstream region the classification approach failed. There, the rivers are surrounded by seasonal wetland whose observations are also water returns. Additionally,

the width of the rivers feature larger seasonal changes than in the other regions. This can influence the waveform and RIP significantly. At some points we find peaky returns in the dry season, which can also be found in the wet season in the wetland, whereas the river itself shows near ocean-like waveforms during in the wet season.

In Figure 4 the mean waveform and mean RIP of some classes are shown ( note the different power-axes). The classes displayed are selected to best represent all 20 classes for the upstream and middle region. As can be seen, the shape of the

mean waveform and mean RIP of water classes in the upstream region reappear in the middle region, but not as water classes. In the middle region small lakes have the same signature as the river upstream. For this reason, the two regions were classified separately. The third land class shown for the upstream region has a very distorted mean RIP. In this area not all stacks over land are 'full', i. e. not every single-look recorded returning power. This leads to such distorted RIPs (side note: in another class the distortion is mirrored). All mean waveforms and RIPs are displayed in Appendix A for the interested reader.

In Figure 5, a section of the river network in the upstream region with the results of the classification is shown. The course of the river is well depicted, however, not at every crossing of the satellite track with the river water is identified. At some crossings no water reflection of the river was measured since the river was too narrow. On the other hand, some points classified as water are not close to the given polygon (blue line). However, the topography model (ETOPO1, Amante and Eakins (2009)) shown in the background indicates river valleys in the three circled areas. Therefore, one can assume that the classification is able to

find rivers that are so small (down to 20 m wide) that they are not present in the high resolution river polygon provided by the MRC.

Figure 6 shows the classification for one exemplary track in the upstream region. The measurements classified as water (red dots) line up to a nearly constant water level at all crossings of the satellite track with the river.

## 6.2 Resulting water level

In the entire Mekong Basin we estimate water levels at more than 2000 crossings, which means approximately one measurement every 4 km along the main river (compared to 50 km for Envisat). It is not possible to measure a water level at every crossing between the CryoSat-2 track with a river in the basin. As mentioned before, at some crossings the river is too small so that not in every pass a reliable measurement could be made; some other water levels were discarded during the outlier detection; furthermore, at some crossings the classification failed to identify the water. However, we are still able to retrieve at

least some measurements from rivers as small as 20 m in width. In Figure 7 all measured heights at all dates are presented in a map, which shows well the overall topography of the river network but cannot show smaller details like seasonal variations.

For one track the heights and the classification are displayed in Figure 6 with an inlaid map of the geographic surroundings. In this track four water crossings are found where the two most northern ones are very close together with a difference of the




**Figure 4.** The mean waveforms and RIPs of some selected classes.

water level of 20 cm. There the river meanders under the track which causes two crossings close together. The two southern crossings are two different rivers which explains the large height difference between the two locations close together. It is visible that only few measurements are used to estimate the water level at each crossing. Approximately 180 water levels (or 8%) are even estimated by just one measurement, the majority of those in the upstream region.

5    For crossings with more than one water measurement we can calculate the standard deviation of the measurements used for water level estimation. More than 85% of the water levels have a standard deviation of less than 0.5 m.

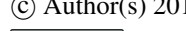





**Figure 5.** An example of the classification. The red dots are classified as water measurements, black are the land classified measurements. The three circles indicate areas where water was detected in river valleys which are not included in the river polygon. The background shows the ETOPO1 topography model.

## 6.3 Validation

The classification is validated twofold: On the one side, we test the repeatability of the classification with a cross validation. On the other side, the different classification in the regions can be compared in the overlap areas. The latter can be used at the same time also to validate the resulting water levels. Additionally, the water levels are validated with respect to the stable

5   seasonal signal using gauge data. We compare these results with the performance of Envisat water levels and CryoSat-2 data extracted with a land-water mask in the same validation.




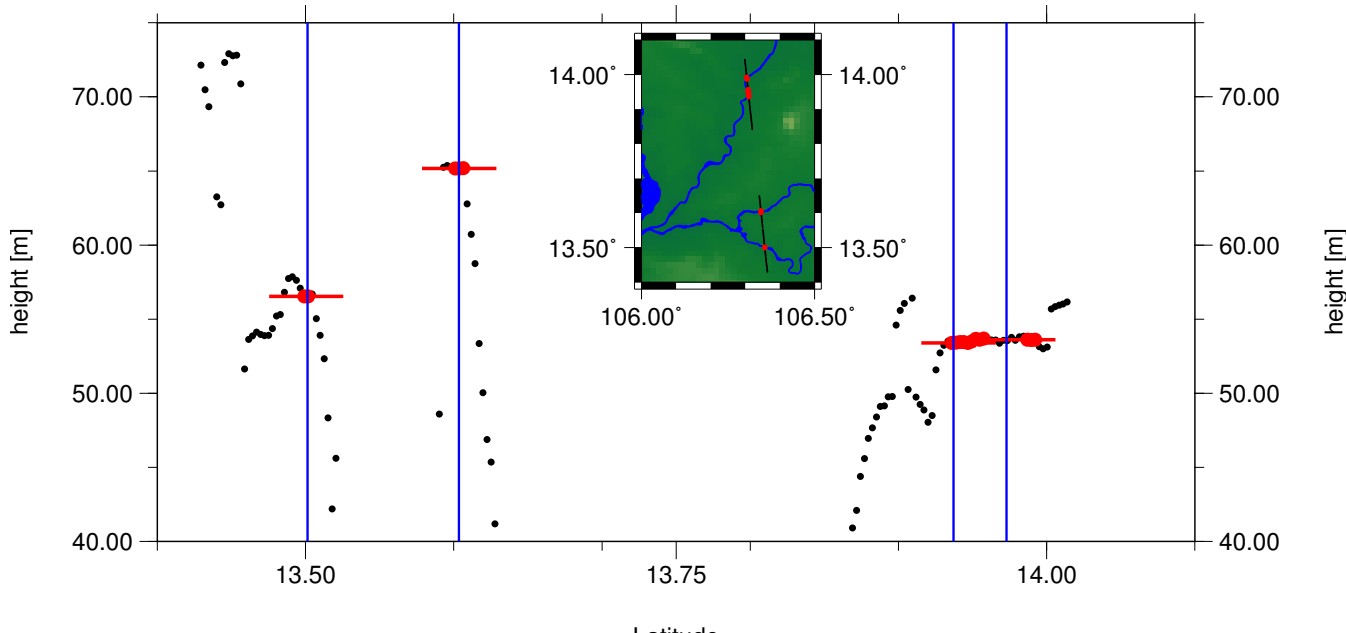

**Figure 6.** Heights along one track which crosses a river at four locations. The inlaid map shows the geographical surroundings with the layout of the rivers. The black dots are all retracked heights with the red dots indicating which measurments were classified as water. The blue vertical lines show the location of the crossing of the track with the river polygone and the horizontal lines are the estimated water level at each crossing.

### 6.3.1 Validation of the Classification

The cross validation of the classification is done for one third of the data. The classes determined before are considered as true values for this validation. The data are split into two equal parts. The first part is again clustered with the k-means algorithm, whereas the second part is classified with the resulting classes. This classification is validated against the "true" classes we found before in the first classification.

Table 2 summarizes the results of the cross validation. Water and non water classes are distinguished. The overall accuracy is 97.9%. This cross validation shows that the classification is stable and does not change with the data subset used for the clustering.

**Table 2.** Result of the cross validation

|  |  | *Classified classes with parts of data* |  |
|---|---|---|---|
|  |  | **Water** | **No water** |
| *Classes with all data* | **Water** | 7321 | 205 |
|  | **No water** | 423 | 22660 |





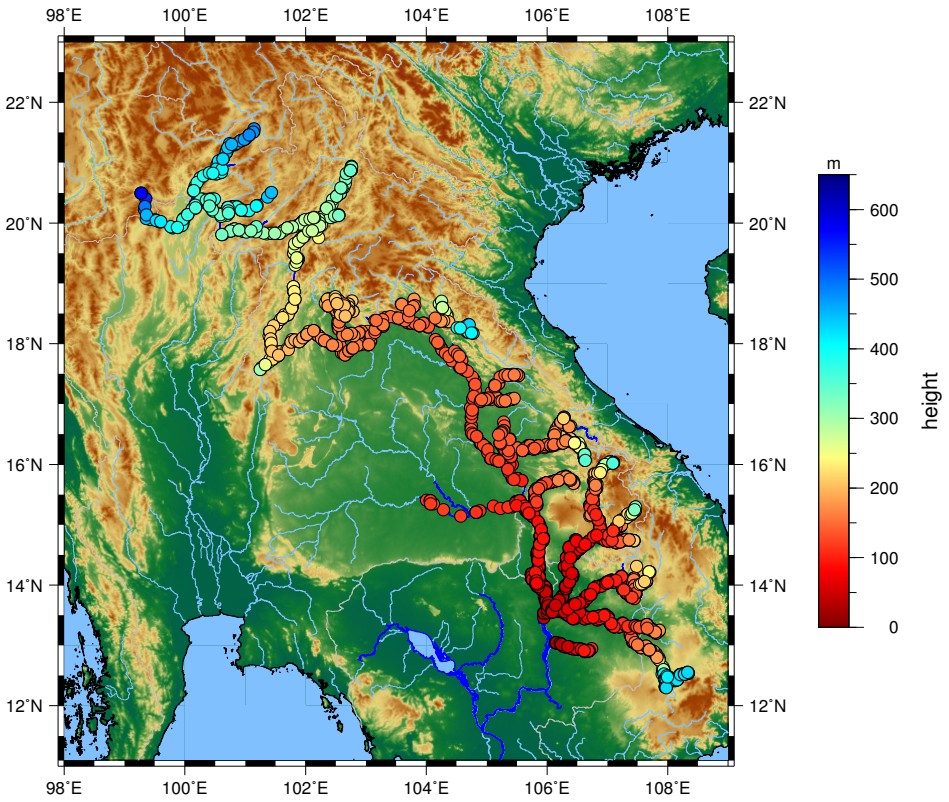

**Figure 7.** Resulting water levels in the Mekong River Basin

### 6.3.2 Validation of Water Levels

Unlike water level time series measured by short-repeat orbit missions, CryoSat-2 measurements cannot be validated against the time series of in situ gauges without reducing the topography as done by Villadsen et al. (2015). The Mekong River and its tributaries have topography that is too complex to allow for reliable reduction. To validate the water levels we use again the nearly one year repeat time of CryoSat-2. We investigate the differences between two subsequent tracks at the same river crossing. A histogram of the differences is shown in Figure 8(a). Table 3 displays the median, mean and standard deviation of these differences for the merged results as well as for the two regions (upstream and middle) separately. The results of the validation are compared to a validation with in situ gauge data, Envisat data and CryoSat-2 data with a land-water-mask. The gauge data provided by the Mekong River Commission for the main river and also some tributaries has a daily temporal resolution (http://ffw.mrcmekong.org/). From Table 3 and Figure 8 one can see that the water level varies up to 50 cm in median from year to year, but some years show much larger differences of up to 4 m. The Envisat data is taken from the DAHITI database (Schwatke et al., 2015) for the main river as well as some tributaries (Boergens et al., 2016b) and has a temporal resolution of up to 35 days. For validation, we take the differences between gauge measurements that are 369 days apart and Envisat measurements where the day of the year is less than 5 days different. The validation of the gauges gives





a measure of how stable the annual signal is in the Mekong Basin. The Envisat observations are the most commonly used data for inland waters with a pulse limited altimeter. We also compare our results to water levels derived from CryoSat-2 by simply averaging measurements inside the land-water-mask (Figure 8(b)). The water levels derived with the land-water mask underwent the same outlier detection as used on the results of the CryoSat-2 classification for better comparability.

The median of the differences of the CryoSat-2 classification results are always better than the Envisat results (see Table 3). Even though, the differences are larger for the upstream region than for the middle region. In the upstream region, the mean difference are nearly equal for CryoSat-2 classification and Envisat results caused by the larger spread of the CryoSat-2 results.

The land-water-mask method lead to comparable good results as the classification along the main stream in the middle region where the river is wide. But in the upstream region with small rivers with a width of 100 m or less the quality deteriorates. The

polygon is given with an accuracy of 50 m which is sufficient for a 1 km wide river but is too inaccurate for 100 m wide rivers. This causes the larger difference in the validation results of the two CryoSat-2 data sets in the upstream region

Additionally, the feature selection of the classification was done mostly with regard to the reflective properties of small water bodies which we find in the upstream region. This explains the better classification results in the upstream region compared to the middle region.

**6.3.3    Validation in the overlap regions**

The overlap between the two regions, upstream and middle described in subsection 5.3, can be used for validation of the classification and height determination.

Theoretically, the land-water classification and the resulting water levels should be identical in the overlap between the two regions. Unfortunately, this is not the case for all points. Overall, at only 67 river crossings the water levels are estimated in

both regions. At these 67 points it is possible to evaluate the differences of the two water levels. Out of these, in 45 cases, or 67%, the differences are below 15 cm where we consider them equal given the accuracy of river altimeter measurements. At the same time, the largest difference between two water levels at the same location is 17 m. At the crossings where the difference is larger than 15 cm it has to be decided which water level is taken for the final data set (see subsection 5.3). In 17 cases the water level of the upstream region and in 5 cases the water level of the middle regions was chosen. We found that the decision

which of the water levels should be taken has a spatial dependency. Towards the upstream border of the overlap region the results of the upstream classification are more likely to be taken, and vice versa for the middle region. Something similar can be observed for those crossings in the overlap region which have only in one of the two data sets water level estimations, we find more valid upstream observations towards the border to the upstream region and more middle stream observations towards the middle stream region. All this together justifies the separation of the classification into the different regions.

**7    Conclusions**

We demonstrate in this study the possibilities of classifying CryoSat-2 SAR data in the Mekong River Basin and using this classification for water level extraction. The classification uses features derived not only from the waveform but also from the





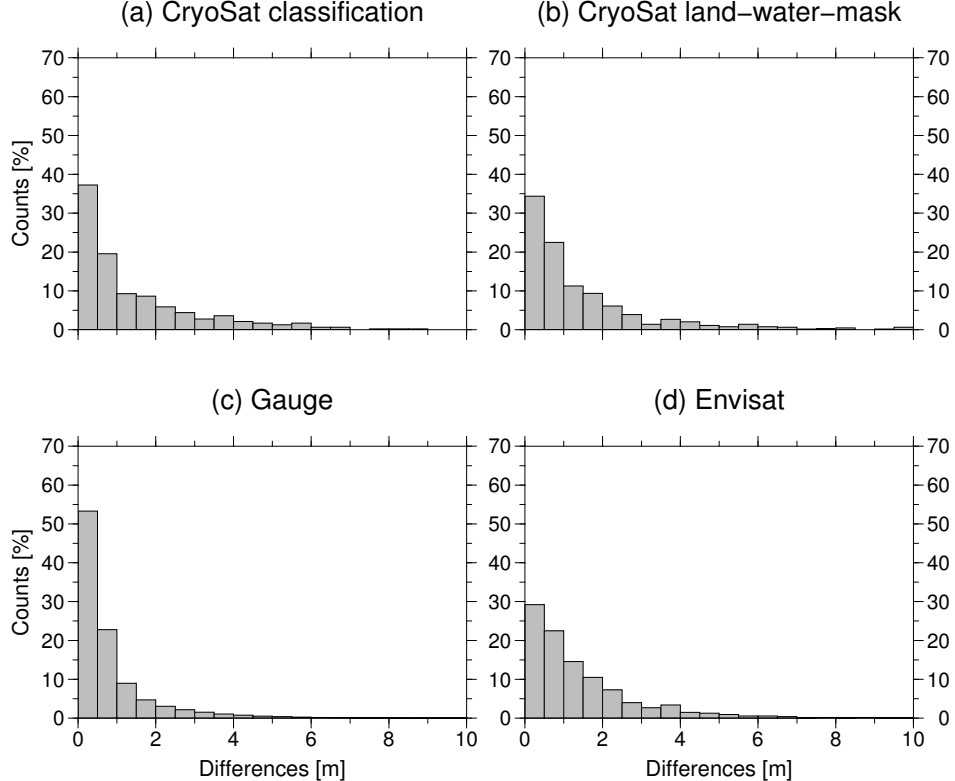

**Figure 8.** Histogram of the differences of height measurements 369 days apart for CryoSat-2 water levels with the classification, CryoSat-2 water levels inside land-water-mask, gauge water level, and Envisat water level.

RIP. The RIP contains more information about the reflecting surface than the waveform on its own can provide. This improves the classification and allowes us to identify even very small rivers with a width as small as 20 m. In fact, the classification works better on smaller rivers than wider rivers. The cross validation of the classification shows that it is stable and repeatable. However, we were not able to use this classification to isolate the river in the downstream region where the Mekong River is

5    surrounded by seasonal wetlands.

The classification in water and land measurements is used to derive water levels at the crossings of the CryoSat-2 track with a river in the whole basin. Overall, more than 2000 water levels are measured, after outlier detection. However, it is not possible to derive at every crossing a water level. The altimeter is not able to measure a water return at every possible river crossing due to too small rivers or too disturbed returns. Additionally, some measured water levels are discarded in the outlier detection.

10    The water levels are validated using the near yearly return time of CryoSat-2 and the very stable annual signal in the basin. This validation is compared to the same validation done on Envisat water levels, gauge measurements and using a land-water-mask on CryoSat-2 data. Especially, for small rivers in the upstream region the classification improves the water level





**Table 3.** Analysis of the differences of height measurements 369 days apart for the whole study area, only the upstream region, and only the middle stream region.

|  | Median [m] | Mean [m] | Standard deviation [m] |
|---|---|---|---|
| Whole study area | | | |
| CryoSat-2 classification | 0.76 | 1.43 | 1.59 |
| CryoSat-2 land-water-mask | 0.83 | 1.86 | 4.55 |
| Gauge | 0.45 | 0.82 | 1.09 |
| Envisat | 0.96 | 1.42 | 1.44 |
| Middle region | | | |
| CryoSat-2 classification | 0.76 | 1.15 | 1.10 |
| CryoSat-2 land-water-mask | 0.84 | 1.55 | 1.87 |
| Gauge | 0.54 | 1.00 | 1.14 |
| Envisat | 0.81 | 1.26 | 1.26 |
| Upstream region | | | |
| CryoSat-2 classification | 0.79 | 1.54 | 1.70 |
| CryoSat-2 land-water-mask | 0.85 | 2.00 | 5.44 |
| Gauge | 0.42 | 0.72 | 1.05 |
| Envisat | 1.01 | 1.46 | 1.49 |

determination compared to the use of a land-water-mask. Compared to Envisat water levels the CryoSat-2 water levels are of higher quality in the whole river basin due to the smaller footprint of the SAR compared to pulse limited altimeter on Envisat.

The resulting water levels of this study will be used in a combination with other altimetric water levels following the ideas of Boergens et al. (2016a) to build basin wide multi-mission water level time series. With CryoSat-2 data we will be able to

5 significantly improve the spatial resolution of the water level observations and to better close the data gap between the end of the Envisat mission and the launch of the SARAL mission. With the launch of the Sentinel-3 satellite in February 2016 SAR altimetry data with a short repeat time is available. When the full stack data are publicly available the same classification of the data for water level retrieval can be hopefully used.

## Appendix A:  Mean waveforms and RIPs

10 *Author contributions.* EB developed the method, conducted the data analysis and wrote the majority of the paper. KN helped with the development of the method and validation. OBA and DD contributed to the discussion of the method and results and writing the manuscript. FS supervised the research and contributed to manuscript writing and organization.





**Figure A1.** Upstream region mean waveforms. Water classes: 0, 4, 13

*Competing interests.* The authors declare that they have no conflict of interest.

*Acknowledgements.* We want to thank the ESA GPOD team who provided us with the full data set through the SARvatore database and who were very helpful with questions regarding the data. This work was supported by the German Research Foundation (DFG) by the TUM International Graduate School of Science and Engineering (IGSSE) and through fund SE 1916/4-1.





**Figure A1.** Upstream region mean RIPs. Water classes: 0, 4, 13

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





**Figure A1.** Middle region mean waveforms. Water classes: 2, 5, 6, 8, 11, 12, 14

Berry, P. A. M., Garlick, J. D., Freeman, J. A., and Mathers, E. L.: Global inland water monitoring from multi-mission altimetry, Geophysical Research Letters, 32, 1–4, doi:10.1029/2005GL022814, 2005.

Biancamaria, S., Frappart, F., Leleu, A.-S., Marieu, V., Blumstein, D., Desjonquères, J.-D., Boy, F., Sottolichio, A., and Valle-Levinson, A.: Satellite radar altimetry water elevations performance over a 200m wide river: Evaluation over the Garonne River, Advances in Space Research, 59, 1–19, doi:10.1016/j.asr.2016.10.008, 2016.

Birkett, C. M.: Contribution of the TOPEX NASA radar altimeter to the global monitoring of large rivers and wetlands, Water Resources Research, 34, 1223–1239, doi:10.1029/98WR00124, 1998.

Boergens, E., Buhl, S., Dettmering, D., Klüppelberg, C., and Seitz, F.: Combination of multi-mission altimetry data along the Mekong River with spatio-temporal kriging, Journal of Geodesy, doi:10.1007/s00190-016-0980-z, 2016a.







**Figure A1.** Middle region mean RIPs. Water classes: 2, 5, 6, 8, 11, 12, 14

Boergens, E., Dettmering, D., Schwatke, C., and Seitz, F.: Treating the Hooking Effect in satellite altimetry data: A case study along the Mekong River and its tributaries, Remote Sensing, pp. 1–22, doi:10.3390/rs8020091, 2016b.

Center, G. R. D.: Long-term mean monthly discharges and annual characteristics of GRDC stations, Tech. rep., Federal Institute of Hydrology, 2013.

5  Commision, M. R.: Annual Mekong Flood Report 2008, Tech. rep., 2009.

Commission, M. R., ed.: Overview of the Hydrology of the Mekong Basin, Mekong River Commission, 2005.

Cullen, R. A. and Wingham, D. J.: CryoSat Level 1b Processing Algorithms and Simulation Results, Geoscience and Remote Sensing Symposium, 2002. IGARSS '02, 3, 1762–1764, 2002.




Desai, S., Chander, S., Ganguly, D., Chauhan, P., Lele, P., and James, M.: Waveform Classification and Water-Land Transition over the Brahmaputra River using SARAL/AltiKa & Jason-2 Altimeter., Indian Soc. Remote Sens., 43, 475–485, 2015.

Dettmering, D., Schwatke, C., Boergens, E., and Seitz, F.: Potential of ENVISAT Radar Altimetry for Water Level Monitoring in the Pantanal Wetland, Remote Sensing, 8, 596, doi:10.3390/rs8070596, 2016.

Doane, D. P.: Aesthetic frequency classifications, The American Statistician, 30, 181–183, doi:10.1080/00031305.1976.10479172, 1976.

ESA: CryoSat-2 Geographical Mode Mask, 2016.

Frappart, F., Do Minh, K., L'Hermitte, J., Cazenave, A., Ramillien, G., Le Toan, T., and Mognard-Campbell, N.: Water volume change in the lower {{}M{}}ekong from satellite altimetry and imagery data, Geophysical Journal International, 167, 570–584, doi:10.1111/j.1365-246X.2006.03184.x, 2006.

Gommenginger, C., Thibaut, P., Fenoglio-Marc, L., Quartly, G., Deng, X., Gómez-Enri, J., Challenor, P., and Gao, Y.: Retracking altimeter waveforms near the coasts, in: Coastal altimetry, edited by Benveniste, J., Cipollini, P., Kostianoy, A. G., and Vignudelli, S., pp. 61–101, Springer, doi:10.1007/978-3-642-12796-0, 2011.

Göttl, F., Dettmering, D., Müller, F. L., and Schwatke, C.: Lake level estimation based on CryoSat-2 SAR altimetry and multi-looked waveform classification, Remote Sensing, 8, 1–16, doi:10.3390/rs8110885, 2016.

Kleinherenbrink, M., Lindenbergh, R. C., and Ditmar, P. G.: Monitoring of lake level changes on the Tibetan Plateau and Tian Shan by retracking Cryosat SARIn waveforms, Journal of Hydrology, 521, 119–131, doi:http://dx.doi.org/10.1016/j.jhydrol.2014.11.063, 2015.

Laxon, S. W.: Sea-Ice Altimeter Processing Scheme at the EODC, International Journal of Remote Sensing, 15, 915–924, doi:10.1080/01431169408954124, 1994.

MacQueen, J.: Some methods for classification and analysis of multivariate observations, Proceedings of the Fifth Berkeley Symposium on 20   Mathematical Statistics and Probability, 1, 281–297, doi:citeulike-article-id:6083430, 1967.

Nielsen, K., Stenseng, L., Andersen, O. B., and Villadsen, H.: Validation of CryoSat-2 SAR mode based lake levels, Remote Sensing of Environment, 171, 162–170, doi:10.1016/j.rse.2015.10.023, 2015.

Raney, R. K.: The delay/doppler radar altimeter, IEEE Transactions on Geoscience and Remote Sensing, 36, 1578–1588, doi:10.1109/36.718861, 1998.

Santos da Silva, J., Calmant, S., Seyler, F., Rotunno Filho, O. C., Cochonneau, G., and Mansur, W. J.: Water levels in the {A}mazon basin derived from the {ERS}- 2 and {ENVISAT} radar altimetry missions, Remote Sensing of Environment, 114, 2160–2181, doi:10.1016/j.rse.2010.04.020, 2010.

Schwatke, C., Dettmering, D., Bosch, W., and Seitz, F.: DAHITI - an innovative approach for estimating water level time series over inland waters using multi-mission satellite altimetry, Hydrol. Earth Syst. Sci, 19, 4345–4364, 2015.

Sturges, H.: The choice of a class interval, Journal of the American Statistical Association, 21, 65–66, doi:10.2307/2965501, 1926.

Villadsen, H., Andersen, O. B., Stenseng, L., Nielsen, K., and Knudsen, P.: CryoSat-2 altimetry for river level monitoring — Evaluation in the Ganges–Brahmaputra River basin, Remote Sensing of Environment, 168, 80–89, doi:10.1016/j.rse.2015.05.025, 2015.

Villadsen, H., Deng, X., Andersen, O. B., Stenseng, L., Nielsen, K., and Knudsen, P.: Improved inland water levels from SAR altimetry using novel empirical and physical retrackers, Journal of Hydrology, 537, 234–247, doi:10.1016/j.jhydrol.2016.03.051, 2016.

Wingham, D. J., Francis, C. R., Baker, S., Bouzinac, C., Brockley, D., Cullen, R., de Chateau-Thierry, P., Laxon, S. W., Mallow, U., Mavro-cordatos, C., Phalippou, L., Ratier, G., Rey, L., Rostan, F., Viau, P., and Wallis, D. W.: CryoSat: A mission to determine the fluctuations in Earth's land and marine ice fields, Advances in Space Research, 37, 841–871, doi:10.1016/j.asr.2005.07.027, 2006.