# Peer review of "Water levels of the Mekong River Basin based on CryoSat-2 SAR data classification"

_Hydrology and Earth System Sciences, 2017_

## Referee Comment (RC1) · Anonymous Referee #1 · 19 Jun 2017

Summary

This paper presents a new method for classifying satellite SAR-altimetry returns over inland water targets. The conventional approach is to select inland water returns from the dataset using static or dynamic water-land masks. This paper presents an alternative methodology, in which properties of the signal itself are used to determine whether the signal is returned by inland water or not. Such a method is useful for the inland water altimetry community, because time-consuming water-land masking steps may be saved and because it may provide new ways of filtering outliers from the altimetric record. The study would benefit from a more systematic comparison of the new approach developed here and the conventional approach using land-water masks.

Review Comments

1. p. 2, line 10ff: The authors state that the/some land-water-masks are "constant over time" and therefore neglect seasonal variations of the water extent. Dynamic water masks can be derived fairly easy from, for example, Sentinel-1 SAR imagery or, cloud cover permitting, from Landsat or Sentinel-2 optical imagery (e.g. NDVI or NDWI). Furthermore (p. 2, line 21), the authors state that their method can "overcome the problems and limitations of land-water-masks". It seems, however, that especially highly variable seasonal water extents hinder the application of the developed classification in the downstream parts of the Mekong (p. 10, line 4 ff: "the width of the river feature larger seasonal changes than in the other regions. This can influence the waveform and RIP significantly") In this case, the introduction shouldn't mention this as an advantage of the developed method.

2. Validation of Water Levels: Besides using the differences between observations 369 days apart, which only provides an indirect way of validating the observed heights, a direct validation against in situ water levels still should be possible. At least for CryoSat-2 observations in proximity of in situ stations, where simple assumptions about river slope can hold true.

3. It would be interesting to compare results to the conventional approach of using a river mask, and simply filter points with that. This has been done by the authors, but only mentioned in the manuscript briefly for the validation. So how many water observations do you get from the river mask vs. the classification, how many outliers, how good is the fit for an actual validation against in situ data as suggested above? The question is whether there is a clear benefit of the classification approach over the conventional river masking.

4. P7L20ff: It appears that, for the validation of the classification, and also in order to find out which of the kmeans clusters represent water (P10L2ff), a water mask is still required... So maybe the selling point for this methodology is not so much that it can operate without land-water-masks but rather that it can improve outlier filtering?
5. P7L24ff: Apparently, the region of interest has to be divided into subregions prior to the application of the method because it is "too diverse in the reflectivity properties of water bodies". How would this play out for a routine application of this method in a new basin, what is the operational procedure to slice the region up into appropriate sub-regions?

6. P15L8ff: It is probably difficult to get to any general conclusion regarding the relative performance of the classification and the masking approaches, as, obviously, the performance of the masking approach will depend on the quality of the mask. Lower performance in the upstream region may simply indicate that the mask is less reliable there. One could even think of a reversed sales argument here, and argue that in regions with narrow rivers SAR altimetry offers a tool to map water surfaces that are too small to be reliably resolved by Landsat/Sentinel SAR imagery.

Details

1. p. 2, line 14 ff: Can we really say that a 30m resolution is insufficient? In the reviewer's experience, such precise water masks often can be buffered (i.e. enlarged) to obtain a higher number of water level measurements. This is likely linked to the size of the footprint of the altimeter, which also in SAR mode is much larger than the resolution of the water mask.

2. The classification of rivers into large, smaller and small is inconvenient and easily misunderstood. It may be better to operate with classes A,B, and C or similar and just define breaks between the classes.

3. P1L14: "smaller upstream regions" should probably be "upstream regions with smaller water bodies".

4. Some in-text citations are messed up (e.g. P1L17, P3L9)

5. P2L9: Please add a few arguments explaining why the dense spatial distribution is an advantage, esp. for rivers.

6. P3L6: "gauges" should probably be "gorges"

7. P3L17-18: Sentence repeated from above.

8. Fig 1: Map legend entries have different symbology from shown layers

9. P7L15: delete "a" before "several"

---

## Referee Comment (RC2) · Anonymous Referee #2 · 24 Jun 2017

The authors use Cryosat-2 SAR data for inferring water levels over the Mekong river system. As with the earlier pulse-limited radars, applying SAR for inland water bodies requires a (static or time-variable) land-water mask for separating water echoes from land returns, that is usually derived from optical and/or radar remote sensing. In this article, the authors suggest to derive the mask from the SAR Delay-Doppler stacks itself, using a standard classification method. It must be noted that Cryosat-2 levels, due to the satellite's unusual 'repeat' orbit, are difficult to validate.

SAR promises to enable water level measurements for rivers of width down to few 100 m or even less, and the availability of river masks poses a challenge, in particular when seasonal inundation is present. The work is timely and touches upon a relevant topic. However, the scientific hypothesis and the paper's objective are not well-described and

it is difficult to read. In large parts, the article is written in an explorative style: the authors apply a certain sequence of approaches and report about success, but it is not explained why these particular approaches are chosen nor are systematic tests provided. This applies to the classification approach (k-means), and the same goes for the features chosen to be used for classification. The authors jump back and forth with approach and validation. I would suggest a more systematic writing: data, method, results, validation, interpretation. Also, it would be helpful to have 1) a flowchart for the approach including region subdivision, classification, retracking, and outlier removal , and 2) a flowchart for the validations.

Other remarks

Page 2 lines 24-30: Here, the authors somehow suggest the range-integrated power, RIP, provides a kind of independent observation that is not available for other altimeters. Although it is true in a literal sense I find this slightly misleading: In fact, both the RIP and the SAR waveforms are derived from the stacks (Fig. 2) which contains the primary observable. In fact all the features they used can be interpreted as properties of the stack matrix. This leads to the question whether the authors actually use the multi-look stacks for focusing on the rivers when deriving water levels? This should be answered in section 3.

In my printout of figure 1, the upstream / mid-stream / downstream mask hachures are not shown as indicated in the inset.

The authors argue that for the smaller rivers of the Mekong basin no reliable land-water mask is available. But there are definitely regions of the world where very precise land-water masks are available (e.g. all EU) – why not testing the classification approach properly for such a region? If this paper is meant to provide a new method, it would be perfectly ok to add comparisons in a test region outside the Mekong.

Eq. (1): the terminology is awkward. Peakiness pwf is not a vector, why is the symbol bold? Max(wf) should read max-over-I wfi

Can the authors please provide an example for the RIP asymmetry parameter caused by a realistic river sloped, assuming the satellite flies along the river (from 246 single looks)?

From the subdivision of the overall basin in three regions, the authors find that in the overlap regions the classification (and thus water levels) does not always agree. That's worrysome but may be expected given the approach. But what is the implication for applications?

After classification, the water surface is identified in retracked levels through searching for a horizontal (or, I guess, sloped) line. Outliers at the margins should tell about the misclassification in the first step. Is it possible to quantify this, e.g. in % of identified water surface from k-means vs. the straight-line in water levels?

k-means can be seen as a special case of maximum likelihood classification under assuming Gaussian distribution and spherical clusters – the distance measure is not weighted in the original method. But weighted k-means may be appropriate whenever features have different geometric meaning and/or units. Are all features equally weighted, does this really makes sense?

7 Conclusions: 'We demonstrate in this study the possibilities of classifying CryoSat-2 SAR data in the Mekong river basin and using this classification for water level extraction'. This statement carries no information at all – please be concise and provide real conclusions (the above is not even a summary).

Eq. (3) While (3) and (4) may have been 'derived' from the OCOG retracker, they represent standard statistical measures. I find it misleading to refer to the OCOG in this respect, which simply makes use of the same statistical moments.

There are some thresholds chosen for the outlier selection based on the near-annual repeat of C-2, height difference of 7 m, 10km / 30day spacing in the second step. It is said these are based on a conservative approach, but more should be provided on

how robust the overall results are with respect to these thresholds.

Appendix: All four figures have the label A1?

---

## Author Comment (AC1) · 2 Aug 2017

Summary

This paper presents a new method for classifying satellite SAR-altimetry returns over inland water targets. The conventional approach is to select inland water returns from the dataset using static or dynamic water-land masks. This paper presents an alternative methodology, in which properties of the signal itself are used to determine whether the signal is returned by inland water or not. Such a method is useful for the inland water altimetry community, because time-consuming water-land masking steps may be saved and because it may provide new ways of filtering outliers from the altimetric record. The study would benefit from a more systematic comparison of the new approach developed here and the conventional approach using land-water masks.

Review Comments

1. p. 2, line 10ff: The authors state that the/some land-water-masks are "constant over time" and therefore neglect seasonal variations of the water extent. Dynamic water masks can be derived fairly easy from, for example, Sentinel-1 SAR imagery or, cloud cover permitting, from Landsat or Sentinel-2 optical imagery (e.g. NDVI or NDWI).

Furthermore (p. 2, line 21), the authors state that their method can "overcome the problems and limitations of land-water-masks".

Thank you for this comment. You are right that dynamic water masks are more or less easy to derive nowadays but with two limitations: The first is the mentioned cloud cover for optical sensors. In regions with strong weather seasonality like the Mekong with the monsoon, during the high water time nearly no images are available due to the rain clouds. The second limitation regards the SAR images, which have with Sentinel-1 a high enough spatial resolution, but are only available since 2014.

We added in the text:

*Extracting dynamic land-water-masks from optical or SAR remote sensing images is difficult in the study area since cloud-free optical data is only available during the dry season with low water level. Moreover, SAR images with sufficient spatial resolution are only available from 2014 on with the launch of Sentinel-1.*

And changed the sentence to:

*To be independent from the accuracy and availability of land-water-masks,....*

It seems, however, that especially highly variable seasonal water extents hinder the application of the developed classification in the downstream parts of the Mekong (p. 10, line 4 ff: "the width of the river feature larger seasonal changes than in the other regions. This can influence the waveform and RIP significantly") In this case, the introduction shouldn't mention this as an advantage of the developed method.

The main problem of the downstream region is not the changing features but the water areas besides the river.  Any classification algorithm will probably meet a problem discerning between water that belongs to the river and water that belongs to a pond or paddy field. We think that our approach would still work if only the features change between seasons as we then could define more water classes (for dry and wet season). But as it is wetlands resemble dry season river

returns this is not possible. We deleted the reference to the changing features in the text and changed it to:

*There, the rivers are surrounded by seasonal wetland whose observations are also water returns, which the classification algorithm cannot discern.*

2. Validation of Water Levels: Besides using the differences between observations 369 days apart, which only provides an indirect way of validating the observed heights, a direct validation against in situ water levels still should be possible. At least for CryoSat-2 observations in proximity of in situ stations, where simple assumptions about river slope can hold true.

Unfortunately, comparing absolute water levels with the gauge is not possible, as they do not have an absolute height value but record only relative level variations. Therefore, only relative changes derived by Cryosat-2 could be validated against the gauge. The second problem arising is that the gauge time series ends with 2012, whereas we do not have CryoSat-2 before April 2011 and then only for part of the region (see figure 2). Even with a margin of 30km around the gauges we get to little CryoSat data for validation against the gauge. To compute a mean and RMSE from 5 overlapping points would not be statistically valid (and this would be the best case we found).

We added at the beginning of the validation section:

*Besides this, the temporal overlap between the CryoSat-2 data and the gauge data is only about 1.5 years or even less (April 2011 until December 2012)*

3. It would be interesting to compare results to the conventional approach of using a river mask, and simply filter points with that. This has been done by the authors, but only mentioned in the manuscript briefly for the validation. So how many water observations do you get from the river mask vs. the classification, how many outliers, how good is the fit for an actual validation against in

situ data as suggested above? The question is whether there is a clear benefit of the classification approach over the conventional river masking.

We think that the benefit of the classification over the mask approach is apparent from Table 4. However, it is true, that we do not mention in the paper the number of water levels gained by both methods. For this we have to separate the two regions. In the upstream region the classification leads to even more valid heights than the mask approach (see table below). And then more of the heights of the mask approach are dismissed in the outlier detection. For the middle stream region the mask approach leads to more water levels but again more are dismissed in the outlier detection. See table below for the exact numbers.

| | Classification | Mask |
| --- | --- | --- |
| Upstream region | | |
| Before outlier detection | 1740 | 1646 |
| After outlier detection | 1703 | 1534 |
| Middle region | | |
| Before outlier detection | 529 | 1417 |
| After outlier detection | 516 | 1364 |

A validation against gauges is not possible as explained above.

We added in the text:

*In the middle region along the main river the land-water-mask and the classification approach yield comparable results in the validation. However, in absolute numbers of observations the land-water-mask approach produce more water levels but with a higher amount of outliers.*

And

*In the upstream region the water levels of the classification approach are superior over those of the land-water-mask approach as well in terms of validation results and absolute numbers of valid observations. For both regions the number of outliers is much larger for the mask than for the classification approach.*

4. P7L20ff: It appears that, for the validation of the classification, and also in order to find out which of the kmeans clusters represent water (P10L2ff), a water mask is still required... So maybe the selling point for this methodology is not so much that it can operate without land-water masks but rather that it can improve outlier filtering?

We do not depend on a land water mask for the validation of the resulting heights. It is only used for the comparison with the land-water-mask approach.

For the k-means cluster identification we only need the approximate location of the river which could also be the center line of the river. We searched for the classes with all points in the vicinity of the approximate location of the river and not only inside the mask. Therefore, we could do this with an even less accurate mask for regions where only the global land-water-masks are available. Additionally, we changed the text to:

*This was done by visual inspection of the mean waveform and RIP for each class and the locations of the observations in each class related to the approximate location of the river known from the land-water-mask (see \autoref{sec:data}).*

5. P7L24ff: Apparently, the region of interest has to be divided into subregions prior to the application of the method because it is "too diverse in the reflectivity properties of water bodies". How would this play out for a routine application of this method in a new basin, what is the operational procedure to slice the region up into appropriate sub-regions?

We used the ETOPO1 topographical model of the region which is globally available (Probably any topography model could be used for this). For each grid cell of the model we calculated a roughness index based on the standard deviation of the heights in a 10km radius. This roughness coefficient and the absolute height were used to define the subregions. Afterwards the borders of the subregions were extended with a margin for the overlap which also smoothed out the edges. For a new basin the same procedure could be employed but the thresholds for roughness and

height might be different. We have not tested this for other basins and it is also highly possible that it is not necessary for every basin to subdivided regions.

We added in the text of section 2:

*The regions are determined by the roughness of a topography model and the absolute height. Afterwards a margin around each subregion allows for an overlap.*

6. P15L8ff: It is probably difficult to get to any general conclusion regarding the relative performance of the classification and the masking approaches, as, obviously, the performance of the masking approach will depend on the quality of the mask. Lower performance in the upstream region may simply indicate that the mask is less reliable there. One could even think of a reversed sales argument here, and argue that in regions with narrow rivers SAR altimetry offers a tool to map water surfaces that are too small to be reliably resolved by Landsat/Sentinel SAR imagery.

That is very true, thank you for the suggestion. We added in the text:

*This shows the opportunity SAR altimetry provides for rivers too small to be reliably identified in optical (e.g. Landsat) or SAR (e.g. Sentinel-1) images. As already shown in \autoref{sec:classification} and \autoref{map_class} the classification of SAR altimetry identifies even rivers which are not visible in the land-water-mask derived from aerial images.*

Details

1. p. 2, line 14 ff: Can we really say that a 30m resolution is insufficient? In the reviewer's experience, such precise water masks often can be buffered (i.e. enlarged) to obtain a higher number of water level measurements. This is likely linked to the size of the footprint of the altimeter, which also in SAR mode is much larger than the resolution of the water mask.

30m is indeed very precise for a land-water-mask, but when buffering the mask to obtain more altimeter measurements one also might (and probably will) get measurements which are not

water reflections. The mask is also not time variable, which is less a problem for the narrow gorges but for the middle part of the river flood expansions.

We reformulated the sentence to:

*Although, a high accuracy land-water-mask is provided by the Mekong River Commission (\url{http://portal.mrcmekong.org/map_service}) which has an accuracy of 30\,m, this accuracy might not be sufficient for small and smaller rivers. Additionally, the mask has no seasonal variations included.*

2. The classification of rivers into large, smaller and small is inconvenient and easily misunderstood. It may be better to operate with classes A, B, and C or similar and just define breaks between the classes.

Thank you for this remark; we agree that it can be confusing for the reader with larger, smaller and small though we defined it at the beginning. However, A, B and C are not easier to understand for a reader who is not reading the whole paper. We decided to rename it to large, medium and small sized rivers.

3. P1L14: "smaller upstream regions" should probably be "upstream regions with smaller water bodies".

We changed the sentence

4. Some in-text citations are messed up (e.g. P1L17, P3L9)

Thank you for this, we fixed the references.

5. P2L9: Please add a few arguments explaining why the dense spatial distribution is an advantage, esp. for rivers.

We added in the text:

*This is especially useful for rivers to better monitor the continuous progression of it. Unlike lakes a river can change its water level rapidly over its course which makes a denser spatial distribution of observations desirable.*

6. P3L6: "gauges" should probably be "gorges"

Changed

7. P3L17-18: Sentence repeated from above.

We removed above the sentence about the mode mask. However, we left the sentence below as we introduce the different mode more thoughtly here.

8. Fig 1: Map legend entries have different symbology from shown layers

There seems to be a problem with this figure, which is also mentioned by the other reviewer. We hope to have fixed this now and that the figure is now visible on all platforms.

9. P7L15: delete "a" before "several"

done

---

## Author Comment (AC2) · 2 Aug 2017

The authors use Cryosat-2 SAR data for inferring water levels over the Mekong river system. As with the earlier pulse-limited radars, applying SAR for inland water bodies requires a (static or time-variable) land-water mask for separating water echoes from land returns, that is usually derived from optical and/or radar remote sensing. In this article, the authors suggest to derive the mask from the SAR Delay-Doppler stacks itself, using a standard classification method. It must be noted that Cryosat-2 levels, due to the satellite's unusual 'repeat' orbit, are difficult to validate. SAR promises to enable water level measurements for rivers of width down to few 100 m or even less, and the availability of river masks poses a challenge, in particular when seasonal inundation is present. The work is timely and touches upon a relevant topic.

However, the scientific hypothesis and the paper's objective are not well-described and it is difficult to read. In large parts, the article is written in an explorative style: the authors apply a certain sequence of approaches and report about success, but it is not explained why these particular approaches are chosen nor are systematic tests provided. This applies to the classification approach (k-means), and the same goes for the features chosen to be used for classification.

Thank you for this concern. More than the presented features were tested. But some were not sensitive for the water classification or redundant with one of the used features (tested with correlation between features). We added in the text at the end of the feature presentation:

*All of these features were chosen due to their sensitivity for the posed problem of water classification and independent from each other. More features were tested but discarded because they were either not sensitive for the classification or highly correlated to one of the used features.*

As for the k-mean algorithm: We needed an unsupervised classification algorithm as we do not have reliable training datasets for a supervised classification. The k-means algorithm is widely used

and was already used for Cryosat-2 waveform classification (Göttl et al.). We also tested the k-mediods algorithm but found no real difference between the results.

We added in the text to the k-mean algorithm in section 4:

*An unsupervised clustering algorithm is used as no reliable training data is available. The unsupervised k-means clustering algorithm is widely used and was already tested for waveform classification in Göttl et al. 2016.*

The authors jump back and forth with approach and validation. I would suggest a more systematic writing: data, method, results, validation, interpretation.

This is a point we discussed among the co-authors before submitting the paper. We found that because of the diverse results it is easier to incorporate the discussion directly into the result and validation section.

Also, it would be helpful to have 1) a flowchart for the approach including region subdivision, classification, retracking, and outlier removal, and 2) a flowchart for the validations.

Thank you for this very good suggestion. We added a flowchart of the approach at the beginning of manuscript. As for the validations, we think that such a flowchart would not be as helpful. The steps of the validation are in parallel and not like the steps of the approach in a hierarchical order. But we added at the beginning of section 6 a table summarizing the different validations done.

Other remarks

Page 2 lines 24-30: Here, the authors somehow suggest the range-integrated power, RIP, provides a kind of independent observation that is not available for other altimeters. Although it is true in a literal sense I find this slightly misleading: In fact, both the RIP and the SAR waveforms are derived from the stacks (Fig. 2) which contains the primary observable. In fact all the features they used

can be interpreted as properties of the stack matrix. This leads to the question whether the authors actually use the multi-look stacks for focusing on the rivers when deriving water levels? This should be answered in section 3.

We changed the text to:

*For the land-water classification a set of features derived from the Cryosat-2 stack data over the intermediate step of the waveform and the RIP is used. The features are summarized in Table 1.*

We are not sure if we understood the second question correctly. Should we have used not the full stack matrix for the waveform which in turn is used for the height determination? In fact we tested using not all single looks of the stack but only those singe looks with high power to calculate the waveform. Another test we did was retracking all single looks and using only those with equal height. Nonetheless, both ways were not improving the results as far as we could validate. Due to the higher computational load this idea was discarded. We state already clearly in section 3 that we use all single look waveforms for the SAR waveform.

In my printout of figure 1, the upstream / mid-stream / downstream mask hachures are not shown as indicated in the inset.

This seems to be a general problem (see review #1). We hope that we solved it now for all platforms and printers.

The authors argue that for the smaller rivers of the Mekong basin no reliable land-water mask is available. But there are definitely regions of the world where very precise land-water masks are available (e.g. all EU) – why not testing the classification approach properly for such a region? If this paper is meant to provide a new method, it would be perfectly ok to add comparisons in a test region outside the Mekong.

It is true, that all rivers inside the EU are very good mapped. But transferring the problem to Europe poses major difficulties. Europa has no nearly as large and diverse river systems as the Mekong which is measured in SAR by Cryosat-2. The Danube River is in SARin and the Po River is according to the mode mask the only larger river measured in SAR. The Po River is not comparable in terms of size and complexity of its surrounding topography to the Mekong River.

As the other reviewer pointed out, the mask available for the Mekong with 30m accuracy is very precise for rivers. Therefore, the comparison between the classification and the land water mask shown in section 6.3.2 is what you ask for.

Eq. (1): the terminology is awkward. Peakiness pwf is not a vector, why is the symbol bold?

Max(wf) should read max-over-I wfi

We changed the symbols

[Figure]

*Figure 1: Symmetry (blue) and off centre (yellow) features for one river crossing (left). The y-axis are the values of the features, x-axis is latitude.*

*The pass (yellow) is ascending and the river (blue) flows towards South-West (right).*

Can the authors please provide an example for the RIP asymmetry parameter caused by a realistic river sloped, assuming the satellite flies along the river (from 246 single looks)?

Please see figure 1 in this document. While approaching the river the surface is sloped towards the satellite and both s and off are positive. s is even decreasing towards the river and then 'jumping' down after the river is passed. How can this be explained? While approaching a water target the overall returned energy increases but the side of the stack towards the water receives relatively more energy which causes the bigger unbalance between the two sides.

The slope of the line of s before and after the crossing are different, after, where the river is sloped away from the satellite, it is flatter than before the crossing.

The slope of the river in this part is between 30 and 40cm/km. We do not think that this example is worth including in the paper.

We also added in the text:

*A positive s indicates a water surface sloped towards the approaching satellite.*

From the subdivision of the overall basin in three regions, the authors find that in the overlap regions the classification (and thus water levels) does not always agree. That's worrysome but may be expected given the approach. But what is the implication for applications?

First of all, the result shows that the classification approach proposed in this study is not a method one to fits it all. The characteristics of water returns differ too much between small and narrow rivers and wide open rivers to be classified with the same parameters. This is something a user has to keep in mind for such a classification; the classification is most successful with a homogeneous target. Still, most heights agree in the overlap considering the accuracy of satellite altimetry over inland waters.

After classification, the water surface is identified in retracked levels through searching for a horizontal (or, I guess, sloped) line. Outliers at the margins should tell about the misclassification in the first step. Is it possible to quantify this, e.g. in % of identified water surface from k-means vs. the straight-line in water levels?

Yes, we search for a horizontal line, but only if enough data points are available (more than 5), which is not that often. We now quantified the number of heights discarded and also looked into the spatial distribution of those discarded. Around 90% of the classified measurements are taken for the height determination. Many of those discarded are apart of the cluster of heights taken for the water level estimation. It seems a reasonable assumption that those measurements are not river reflections but could come from ponds or paddy fields. As we explained in the text another

reason for the outliers could be off nadir effects which occur in SAR data across track. In both cases the classification is not wrong because it classifies correctly water. But we do not want all water but just river water. We do not think that this investigation is interesting enough for the paper.

k-means can be seen as a special case of maximum likelihood classification under assuming Gaussian distribution and spherical clusters – the distance measure is not weighted in the original method. But weighted k-means may be appropriate whenever features have different geometric meaning and/or units. Are all features equally weighted, does this really makes sense?

It does not make sense to use the unweighted k-mean clustering if the units and orders of magnitudes of the features are different or the variance is different. Therefore, we normalized all features before the clustering which also reduced the differences in variance. Still we assume the equal Gaussian distribution for the features. We tested the features for this; all except the maximum power feature passed this test.

We added in the text:

*The k-means algorithm assumes normally distributed features with equal variance, which we ensured by the normalization of the features.*

7 Conclusions: 'We demonstrate in this study the possibilities of classifying CryoSat-2 SAR data in the Mekong river basin and using this classification for water level extraction'. This statement carries no information at all – please be concise and provide real conclusions (the above is not even a summary).

We changed the sentence (also according to suggestions of other reviewer) to:

*We demonstrate in this study the advantage of CryoSat-2 SAR altimetry data for measuring rivers which are identified by a classification, which is independent of a precise land-water-mask.*

Eq. (3) While (3) and (4) may have been 'derived' from the OCOG retracker, they represent standard statistical measures. I find it misleading to refer to the OCOG in this respect, which simply makes use of the same statistical moments.

We removed the reference to the OCOG.

There are some thresholds chosen for the outlier selection based on the near-annual repeat of C-2, height difference of 7 m, 10km / 30day spacing in the second step. It is said these are based on a conservative approach, but more should be provided on how robust the overall results are with respect to these thresholds.

The time of 30 days has only a small impact, if we take anything larger than 30 days it does not change the results. For 10 days only very few data points are additionally kept but these do not have any influence on the final validation approach.  Changing the distance of 10 km has similar small effects as the time spacing.  But it should be kept in mind, that if the spatial and temporal spacing is too small no comparison can be done and the measurement is not considered an outlier. And the mean height to which the height is compared to is weighted by distance in space and time.

As for changing the 7m the effect is a bit larger.  Unsurprisingly, a value less than 7m results in more outliers and vice versa. Looking at Figure 9, the threshold determines where the tail of the distribution is cut off. As one can see, the number of differences in this region of the histogram is very small; therefore a shift of the threshold is not affecting most of the observations. If one looks at the two regions separately the upstream region has a heavier tail of the histogram and thus the threshold of 7 m has a higher influence.

We added in the text:

*Of the three thresholds used for the outlier detection the difference of 7\,m in between years is the most sensitive for the later result. The time and distant weighted mean in the second part of the outlier detection limits the sensitivity of the other two thresholds.*

Appendix: All four figures have the label A1?

This was an error of the Latex Template we were now able to fix.

---

## Author Comment (AC3) · 2 Aug 2017

[revised manuscript text omitted]

10 better monitor their continuous progression. Unlike lakes, rivers can change their water levels rapidly over their course which makes a denser spatial distribution of observations desirable.

To derive water levels from lakes or rivers it is necessary to identify the water returns of the altimeter. This can be done by applying a land-water-mask such as the mask provided by the World Wildlife Fund (https://www.worldwildlife.org/pages/global-lakes-and-wetlands-database). Such a mask is constant over time, therefore, it neither accounts for the seasonal vari-

15 ations of the water extent nor inter-annually shifting river and lake banks. These masks are usually not accurate enough for narrow rivers where only a few water measurements are available. Extracting dynamic land-water-masks from optical or SAR remote sensing images is difficult in the study area since cloud-free optical data is only available during the dry season with low water level. Moreover, SAR images with sufficient spatial resolution are only available from 2014 on with the launch of Sentinel-1.

20  Although, a high accuracy land-water-mask is provided by the Mekong River Commission ($http://portal.mrcmekong.org/map_service$) which has an accuracy of 30 m, this accuracy might not be sufficient for medium and small sized rivers. Additionally, the mask has no seasonal variations included. In the Mekong River Basin the river width varies between 20 m to more than 2 km. The small rivers with a width of less than 100 m are most of the tributaries and the upstream part of the left river bank side main

[revised manuscript text omitted]

$$w = \frac{\left(\sum_i \mathrm{RIP}_i^2\right)^2}{\sum_i \mathrm{RIP}_i^4}. \tag{3}$$

The off-center feature $off$ describes the deviation of the main reflection from the nadir point. It should be close to zero for measurements of water, whereas land measurements are more disturbed and often show the maximum return in the lobes. We measure the off-center feature $off$ as the difference between the middle look of the RIP and the mean point of the RIP which is calculated with:

$$off = \frac{246}{2} - \frac{\sum_i i\mathrm{RIP}_i^2}{\sum_i \mathrm{RIP}_i^2}. \tag{4}$$

A positive $off$ value indicates that the majority of the returning power was detected before the satellite passed the nadir position, a negative value vice versa.

The last feature is a measure of the symmetry of the RIP $s$. For an ideal smooth water reflection, like a small lake, the RIP should be perfectly symmetrical. However, for a sloped target, as a river is, the reflection depends on the relative orientation between the satellite and the water surface. The reflection is stronger when the satellite looks on a water surface that is sloped towards it. A positive $s$ indicates a water surface sloped towards the approaching satellite. This effect leads to an unsymmetrical RIP. To quantify this, an unsymmetrical exponential function $\overline{\mathrm{RIP}}$ is fitted to the RIP with

$$\overline{\mathrm{RIP}_i} = \begin{cases} a\exp\left(\frac{(i-b)^2}{2c_1^2}\right), & \text{if } i < b \\ a\exp\left(-\frac{(i-b)^2}{2c_2^2}\right), & \text{if } i \geqq b. \end{cases} \tag{5}$$

Here, $a$ is the amplitude of the exponential function, $b$ the look where the function reaches its maximum, and $c_1$ and $c_2$ are the two decay parameters. The *symmetry* feature is then

$$s = c_1 - c_2. \tag{6}$$

**Table 1.** Features used for the classification

| RIP features | Waveform features |
| --- | --- |
| Peakiness: $p_{\mathrm{RIP}}$ | $p_{\mathrm{wf}}$ |
| Standard deviation: $std_{\mathrm{RIP}}$ | Maximum power: $max_{\mathrm{wf}}$ |
| OCOG width: $w$ | Relative position of leading edge |
| Off-center: $off$ | |
| Symmetry: $s$ | |

Figure 4, right hand, displays a RIP with the feature $w$ marked. The off-center feature $off$ is too small to be visible in this example, but the symmetry, or the lack thereof, is clearly shown.

Additional to these eight features, both the whole waveform and the whole RIP are used as features. Each bin is then considered as a single feature. The waveform needs to be shifted so that the leading edge is positioned on the nominal tracking point. Since the features span different orders of magnitude, it is necessary to normalize the feature set. All of these features were chosen according to their sensitivity for the posed problem of water classification and independent from each other. More features were tested but discarded because they were either not sensitive for the classification or highly correlated to one of the used features.

[revised manuscript text omitted]

5    The thresholds for the outlier detection were chosen as a conservative upper bound. It has to be expected to have in average a water level difference of 40 to 60 cm in five days during the rising water season, but it could be as high as 4 or 5 m (Mekong River Commission, 2009). Additionally, some inter-annual changes in the flood season can be expected,  and the slope of the river has to be considered which is in median 30 cm/km for the Mekong River. Of the three thresholds used for the outlier detection the difference of 7 m w.r.t. of the year is the most sensitive for the

10  later result. The time and distance weighted mean in the second part of the outlier detection limits the sensitivity of the other two thresholds.

**5.3  Merging of the overlap regions**

[revised manuscript text omitted]

5    **6.3.1   Validation of the Classification**

The cross validation of the classification is done for one third of the data. The classes determined before are considered as true values for this validation. The data are split into two equal parts. The first part is again clustered with the k-means algorithm, whereas the second part is classified with the resulting classes. This classification is validated against the "true" classes we found before in the first classification.

10    Table 3 summarizes the results of the cross validation. Water and non water classes are distinguished. The overall accuracy is 97.9%. This cross validation shows that the classification is stable and does not change with the data subset used for the clustering.

**Table 3.** Result of the cross validation

| | | Classified classes with parts of data | |
|---|---|---|---|
| | | **Water** | **No water** |
| *Classes with all data* | **Water** | 7321 | 205 |
| | **No water** | 423 | 22660 |

As second possibility for the validation of the classification lies in the water level estimation. For crossings with enough measurements only those points which lie on a straight line are used for the height determination (see section 5). The number

15    of observations discarded should be small, if not zero, for a flawless classification.

**6.3.2   Validation of Water Levels**

Unlike water level time series measured by short-repeat orbit missions, CryoSat-2 measurements cannot be validated against the time series of in situ gauges without reducing the topography as done by Villadsen et al. (2015). The Mekong River and its tributaries have topography that is too complex to allow for reliable reduction. Besides this, the temporal overlap between the

20    CryoSat-2 data and the gauge data is only about 1.5 years or even less (April 2011 until December 2012).

[revised manuscript text omitted]